# Effect of Omega 3 supplementation during pregnancy and lactation on cognitive functions in rat offspring

Hande Nur Onur Öztürk[1]*, Perim Fatma Türker[2]

1 Nutrition and Dietetics Department, Istanbul Gelisim University, Faculty of Health Science, Istanbul, Türkiye, 2 Nutrition and Dietetics Department, Acibadem University, Faculty of Health Science, Istanbul, Türkiye

* hnonur@gelisim.edu.tr

## Abstract

### Aim

This study investigates the effects of omega-3 supplementation starting in the pre-conception period, during the 1st, 2nd, and 3rd weeks of pregnancy, and lactation on the cognitive functions of rat offspring.

### Methods

The study involved 30 mother rats divided into a test group receiving 400 mg/kg/day omega-3 supplementation via oral gavage and a control group with no supplementation. Omega-3 supplementation began at different stages for each group and continued through the lactation period (21 days). Two male pups from each mother, for a total of 60 pups, were tested for cognitive function at 45 days using the Morris Water Maze to assess spatial learning and reference memory.

### Results

Significant differences were found between groups in initial weights, pre-mating weights, pre-birth weights, number of pups, and gestation duration ($p < 0.05$), but not in weight gain during pregnancy ($p > 0.05$). The time to find the platform on test days 1 and 2 was significantly different between groups ($p < 0.05$). Differences in platform-finding times across trials were also significant ($p < 0.05$). However, no significant difference was found in the probe test on day 5 ($p > 0.05$).

### Conclusion

Initiating omega-3 supplementation during the early stages of pregnancy may be more effective for both short-term and long-term memory.

**Data availability statement:** I can share my data.

**Funding:** The author(s) received no specific funding for this work.

**Competing interests:** The authors have declared that no competing interests exist.

## Introduction

From conception to 2 years of age, it is named the first 1000 days of life, critically important for the child's future health [1]. The effects of nutrition on tissues and organs during this period of intense cell division can be observed throughout life [2]. The maternal nutritional condition at the time of conception plays a vital role in influencing embryonic and fetal development. Fetal growth is most vulnerable to the mother's nutritional condition during the rapid placentation phase that occurs in the preimplantation period and the first few weeks. Development of most organs are 3–7 weeks after the last menstrual period, during which any teratogenic effects may occur. The mother's nutritional condition may vary by numerous factors such as genetics, lifestyle, environment, diseases, physiological factors, and exposure to drugs or toxic substances [3]. The preconception period refers to the time before pregnancy. Its importance lies in enabling the mother to prepare herself physiologically and psychologically for gestation. Adequate and balanced nutrition during this period plays a crucial role in reducing complications that may affect the health of the baby [4]. Women who report "prudent" or "health-conscious" eating habits before or during pregnancy may have fewer adverse pregnancy outcomes related to child development [5]. Preconception care is a concept proposed to address maternal health issues and environmental risk factors prior to pregnancy in order to improve pregnancy outcomes. In practice, preconception care strategies are defined as behavioral, biomedical, and social interventions offered to couples before conception, aiming to prevent certain avoidable diseases through improved lifestyle choices [6]. At the extremes of maternal malnutrition, the fetus develops chronic fetal growth restriction to "survive at the expense of growth". In this phenotype, pancreatic growth and development are diminished, leading to lower insulin secretion. Peripheral tissues, like skeletal muscle, show an enhanced ability to absorb glucose, while amino acid use for protein synthesis and cell growth is reduced. Additionally, hepatic insulin resistance and fetal hypoxia develop [7]. Individuals' positive or negative early life experiences can transmit to the following generation through maternal and paternal genes that shape the early life environment. As a result, a cycle of passing down "health capital" occurs from one generation to the next [8].

The Developmental Origins of Health and Disease (DOHaD) hypothesis suggests that exposure to adverse conditions such as malnutrition, stress or infection during the prenatal period may show off long-term or permanent effects on the offspring's health pathway during a sensitive developmental period, referred to as "fetal programming" [9]. Programming is the response of an organism that reacts to the most critical period of development. Because developmental processes follow a particular order, changes by external signals that consist in response to growth or maturation stages are frequently permanent. Therefore, the environment may have a long lasting effect on physiological and life-long health and well-being [10]. Human and animal studies have extensively demonstrated the significance of nutrition in early life in predisposing individuals to non-communicable diseases such as cardiovascular diseases and diabetes in adulthood. It is proposed that environmental elements

during key stages of prenatal and postnatal development can shape developmental pathways and predispose offspring to a higher risk of diseases [11–13].

Epigenetics refers to heritable changes in gene expression that do not involve alterations in the DNA sequence. These changes are regulated by mechanisms such as DNA methylation and histone modifications, which can be influenced by environmental and nutritional factors. While the genome provides the organism's potential, epigenetic mechanisms shape the epigenotype, linking environmental exposures to phenotypic outcomes and playing a key role in developmental processes and long-term health [14]. The maternal nutrition conduces to a fetal "epigenetic signature" that influences individuals' sensibilty to disease risk later in life. Epigenetic modifications in fetal tissue play a critical role in the programming of metabolic diseases through the interaction of the maternal environment with gene function. Such epigenetic modifications can happen through changes by histone modification, deoxyribose nucleic acid (DNA) methylation, and non-coding ribonucleic acid (RNA) [15]. After fertilization, epigenetic signs from the parental gametes are largely erased, preparing the cells to receive cell and tissue specific marks during gastrulation and continuation, creating an especially dynamic phase of epigenetic reprogramming. This initial developmental phase, involving molecular systems that maintain parent-specific methylation in genomic imprints, underscores the post-conception period as crucial for epigenetic influences on fetal programming [16]. Fetal programming that occurs during the intrauterine period of life may be related with maternal genes that affect the fetal phenotype independently of the fetal genome. It is suggested that certain maternal genes can affect the fetal phenotype nevertheless of the transmission of the gene to the phenotype [17]. Researchers have linked an undernourished fetus' adaptations to cardiovascular disease, attributing it to the redistribution of metabolic, hormonal, or cardiac output to protect sensitive organs like the brain and reduce growth to meet necessary nutrient requirements. Changes in human lifestyle or treatments received in adulthood, certain conditions that the fetus is exposed to during early development tend to have lasting effects on the body's structure and function [18–20]. Despite various mechanisms, intrauterine programming that occurs under conditions of fetal undernutrition or overnutrition leads to the development of individual predispositions to Type II diabetes, obesity, metabolic syndrome (MS), and cardiovascular diseases. It shows that the impact of birth weight on blood pressure throughout life is highly complex. Most likely, hypertension is induced by both low birth weight and excessive weight gain beyond the normal limits for the stage of pregnancy [21]. The hypothalamic-pituitary-adrenal (HPA) axis is one of the main fetal organ systems sensitive to the programming effects of adverse early life environments. Regulating the neuropsychiatric stress response, maladaptive programming of the HPA axis may be a crucial mechanism which early life adversity predisposes individuals to poor health in adulthood [22]. It has been shown that stress physiology, particularly glucocorticoid hormones produced by the HPA axis, can affect similar long-term health variations and that epigenetic modifications serve as biological memories linking early life experiences to altered patterns of gene regulation. The focus is on the role of maternal exposure to adverse environments, the importance of early experiences in shaping growth and development, and the significant intergenerational contribution of shared social health trends through developmental plasticity [23].

Brain development is a complex process that begins in the third week of pregnancy and lasts throughout life [24]. The most rapid brain weight increases during early childhood, by age 3, then slows down, finally plateauing around age 10. The individual's brain weight then begins to decrease around the fifth decade of life, reaching a level in old age that is approximately 11% lower than in young adulthood [25]. As an important part of brain development, the assumption that the nervous system has a fixed number of neurons and neural networks is no longer widely accepted [26]. With the advent of modern neurobiological methods, evidence has emerged suggesting that the driving and supporting factors of brain development are the interaction between genetic factors and individual experience. Understanding the origins and development of the brain and behavior hinges on recognizing how genetic and environmental factors contribute to the dynamic and interactive processes that shape the neurobehavioral system's growth and function [24]. In humans, neuron development begins in the beginning of pregnancy, and by the end of the sixth month (second trimester), most neurons have formed. The formation of synapses between neurons is quite rapid until around the age of six. After the age of fourteen,

 

the rate of neuronal renewal, repair, and synapse formation gradually decreases. Today, it is proven that neurons can repair themselves, and renew and that the formation of new neurons, though limited, can continue [26].

Synaptic plasticity refers to the morphological and functional changes in pre-existing synapses of the nervous system in response to internal and external environmental stimuli. These activity-dependent changes are essential for brain development, the establishment of neuronal circuits, and synaptic transmission [27]. It is widely believed that synaptic plasticity underlies the formation of long-term memories. The processes that trigger, regulate, and sustain synaptic plasticity have been uncovered in different organisms, and it is believed that these processes are closely related to those governing long-term memory formation. Therefore, the initiation of synaptic plasticity may involve epigenetic mechanisms associated with long-term memory. Epigenetic changes caused by plasticity are also observed in synaptic plasticity in mammalian models. Form of synaptic plasticity named long-term potentiation (LTP) in which synaptic strength is increased in response to high-frequency synaptic activity [28]. The epigenetic marking of the genome during development is the most definitive example of long-term memory storage in multicellular organisms. Once cells have differentiated, they retain their phenotype. Consequently, if neurons and memory formation utilize evolutionarily conserved mechanisms, it is likely that the nervous system has adapted these epigenetic processes to facilitate the creation of long-term memories [29].

Cognition can be defined as the process in which external stimuli, such as touch, hearing, and vision, are transmitted to the cortex, with responses originating from different areas of the brain. Cognitive function encompasses attention processes such as focusing, shifting, dividing, or maintaining attention during a specific task or in response to a stimulus; executive functions such as planning, organizing thoughts, initiating, inhibiting/suppressing, and emotional control; visual-perceptual abilities such as visual scanning, drawing, and construction; memory through recall and retrieval of visual and auditory information; and language, including expressive verbalization and receptive comprehension [30,31]. Neuropsychological assessments can be categorized according to the cognitive domain being tested. Neuropsychological evaluations can assess cognitive function for the frontal lobe, temporal lobe, parietal lobe, hippocampus, or other brain structures [32]. During fetal development, sufficient amounts of energy, protein, essential fatty acids, and various key micronutrients are required as substances necessary for fetal tissue formation in the central nervous system and as key components in biochemical processes that regulate normal brain development [33].

Fatty acids are found in membrane phospholipids and are important in prenatal growth and development, cognitive and behavioral development, and energy metabolism [34]. The lactation period, like the pregnancy period, is especially important for the differentiation and maturation of organs such as the brain, liver, pancreas, adipose tissue, which regulate glucose homeostasis. Bioactive components in breastmilk, such as hormones, growth factors, antioxidant enzymes, immunoglobulins, and nucleotides participate in metabolic regulation and may moderate growth and development during infancy [35]. Nutrients are crucial not just for physical growth but also for brain chemistry and function. In late fetal and early neonatal stages, regions like the hippocampus, visual and auditory cortices, and striatum experience rapid development through processes like morphogenesis and synaptogenesis, which enable their functionality. For cognitive development, there are specific sensitive periods where environmental factors, such as diet, can have enduring impacts. Neurochemical changes include alterations in neurotransmitter synthesis, receptor synthesis, and neurotransmitter reuptake mechanisms. Neurophysiological changes reflect alterations in metabolism and signal transmission [36].

The omega-3 (n-3) fatty acid precursor alpha-linolenic acid (ALA) is metabolized into eicosapentaenoic acid (EPA) and docosahexaenoic acid (DHA), which are considered the two major n-3 fatty acids. ALA is in plant sources such as leafy greens, plants, vegetable oils, and seeds like flax seed and canola. The finest sources of EPA and DHA are marine fish and seafood. EPA and DHA are related to many important functions such as neural activity, including cell membrane fluidity, neurotransmission, ion channels, enzyme regulation, gene expression, and myelination. DHA supplies about 30% of phosphoglycerides in brain gray matter and is essential for optimal neuronal function. DHA accumulates preferentially in growth cones, astrocytes, synaptosomes, myelin, microsomal, and mitochondrial membranes in brain tissues. Omega-3

fatty acids mediate various key neurotransmitter functions, including serotonergic response, signal transmission, and phospholipid cycling [37].

Due to the rapid brain growth and increased synaptogenesis during the perinatal period and early months, the need for polyunsaturated long-chain fatty acids (LC-PUFAs), DHA, and arachidonic acid (AA) increases. Eicosanoids derived from PUFAs affect cell division, synaptogenesis, and neural cell apoptosis. Just as the brain changes throughout life, the effects of nutrition on cognition may also change [38]. Essential fatty acids are of particular importance in time of intrauterine life, a period when brain development and organogenesis occur within gestation. Fatty acid analysis of total lipids taken from fetal brain tissues has shown that the accumulation of total n-3 and omega-6 (n-6) fatty acids gradually increases during the last three months of pregnancy. In fetal liver tissue, a gradual increase in fatty acids has been noted, reflecting the high demand for these nutrients during intrauterine growth [39]. Omega-3 and omega-6 LCPUFAs are essential for the development of the infant and child's brain, playing key roles in various neuronal functions, including membrane fluidity and gene expression regulation. DHA starts to accumulate in the brain during pregnancy, with a notable increase occurring in the latter half of gestation. Deficiencies and imbalances in LCPUFAs are related with impairments in cognitive and behavioral performance [40]. When babies come into the world, maternal intake of essential fatty acid is the main source of breast milks' AA and DHA. For better neurodevelopment, DHA content of breast milk has been documented. The impact of varying DHA levels in breast milk and the effects of DHA supplementation restricted to the breastfeeding period are still under examination [41]. It has been shown that in time of pregnancy deficiency of n-3 fatty acids is related to visual and behavioral deficits that are irreversible by supplementation after birth. For these reasons, it is important to provide sufficient amounts of n-3 fatty acids to the fetus throughout pregnancy [42]. DHA constitutes 90% of the n-3 PUFAs and approximately 10–20% of the total lipids in the human brain and is associated with positive effects on maternal and infant health. Higher DHA intake is associated with a reduced risk of schizophrenia, bipolar disorder, depression, anxiety, and behavioral disorders, while insufficient DHA levels appear to be a potential risk factor for psychiatric disorders [43].

Although the current study focused on behavioral outcomes, it is important to consider the potential molecular mechanisms that may underlie the cognitive effects observed in response to omega-3 supplementation. Omega-3 fatty acids, particularly docosahexaenoic acid (DHA), have been shown to support neuroplasticity by enhancing synaptic connectivity, dendritic branching, and long-term potentiation (LTP), all of which are essential for learning and memory processes [44,45]. Furthermore, omega-3s are involved in the regulation of synaptogenesis and neurotransmitter function, which may contribute to more efficient neural signaling and cognitive performance [46]. Beyond these immediate neuronal effects, recent research also points to the role of epigenetic modifications, such as DNA methylation and histone acetylation, which can influence gene expression patterns related to brain development and plasticity [47].

Taken together, these mechanisms suggest a comprehensive neurobiological framework in which omega-3 fatty acids modulate brain function not only through structural and functional neural changes but also through long-term gene-environment interactions. Future studies incorporating molecular analyses are needed to validate and expand this theoretical model.

The literature suggests that omega-3 supplementation during pregnancy and lactation is important for the offspring's brain and cognitive development. However, the effects of the timing of supplementation on cognitive function are unknown. This study is an in vivo investigation aimed at exploring the effects of omega-3 supplementation, initiated at different stages of pregnancy (preconception and during pregnancy) and lactation, on cognitive functions in offspring. It is unique in its focus on the impact of the timing of omega-3 supplementation.

The study is designed to examine the effects of n-3 fatty acid supplementation, initiated during the preconception period, the 1st, 2nd, and 3rd weeks of pregnancy, and the lactation period in rats, on cognitive functions in offspring. It is hypothesized that the longer the duration of omega-3 supplementation, the greater the improvement in cognitive functions in the offspring.

## Materials and methods

The study was conducted at the Experimental Medicine Research Laboratory (DETALAB) of Istanbul University-Cerrahpaşa. With the ethical approval granted by the Istanbul University-Cerrahpasa Local Animal Experiments Ethics Committee dated 11.01.2023 and numbered 2022/26, and the Baskent University Medical and Health Sciences Research Ethics Committee approval dated 29.03.2023 and numbered 219259, the animals used in the study were produced at DETALAB under project approval code DA23/05. The study was completed between May and September 2023.

Based on the power analysis, 30 primiparous 8-week-old *wistar-albino* female rats (200-300g) were determined to be needed with an effect size of 0.2 and 80% power. The rats were fed *ad libitum* with standard pellet feed (Optima Feed), with no restriction on water intake, which was refreshed daily throughout the experimental period.

All animals were housed in polypropylene cages under standard conditions with a temperature of 22–24°C and a 12-hour light – 12-hour dark cycle. Prior to mating, each cage contained 2–3 animals, and during mating, each male was paired with 2–3 females in a harem setup.The groups of rats, monitored for pregnancy through weight checks, were identified with a permanent marker on their tails and separated based on their expected delivery dates. Rats thought to be close to giving birth were placed in individual cages to deliver. They remained with their pups until weaning (approximately 21 days). After weaning, the mothers were removed from the cages, and the pups were kept together and allowed to grow to determine their sex. Behavioral testing using the Morris Water Maze was conducted when the offspring reached post-natal day 45, an age considered sufficient for the development of spatial learning and memory abilities. To minimize the potential confounding effects of hormonal fluctuations associated with the estrous cycle, which typically begins between postnatal days 30 and 40 in female rats, only male offspring were included in the experimental procedures [48,49].

Six groups, each consisting of 5 animals, were established. These groups were designated as above (Table 1). As a result of the literature research [11–13,50–59], the daily dose of 400 mg/kg/day (Solgar Omega 3 950 mg EPA+DHA), which is suitable for daily use, was given to mother rats via oral gavage. The control group did not consume any supplements. As the weights of the pregnant and lactating rats changed, they were weighed daily between 12:00 and 12:30 PM (using a Beurer BY 90 scale), and the amount of n-3 fatty acid supplementation was adjusted daily.

The Morris Water Maze (MWM) test, developed by Richard Morris in 1984, is designed as a method to assess spatial memory and navigation. In this test, subjects use spatial cues around an open swimming pool to locate a hidden escape platform starting from a designated point. Spatial learning is assessed through repeated trials, while reference memory is evaluated based on the subject's preference for the platform area even when the platform is absent. The test is largely independent of activity levels or body weight differences, making it ideal for many experimental models [60].

The tank was filled with water at a height sufficient to cover the platform, maintained at 22±2°C. To prevent the pups from seeing the platform, milk powder was used to obscure it. The tank was divided into four imaginary quadrants (north, south, east, west). To help the rats navigate, three types of cues (round, star, and square) were placed on the upper wall of the tank in the north, east, and west directions. The platform's location was kept in the same quadrant during the testing period (the first four days) but was removed during the probe test (the fifth day). The pups were placed into the water from a different quadrant each day.

**Table 1. Workflow of experiment.**

| Group | Group Name | Start of Omega-3 Supplementation | Supplementation Period | Total Duration (days) |
|---|---|---|---|---|
| P | Preconception | One week before pregnancy | Pregnancy+Lactation | 49 days |
| P1 | 1st Week of Pregnancy | First week of pregnancy | Pregnancy+Lactation | 42 days |
| P2 | 2nd Week of Pregnancy | Second week of pregnancy | Pregnancy+Lactation | 35 days |
| P3 | 3rd Week of Pregnancy | Third week of pregnancy | Pregnancy+Lactation | 28 days |
| L | Lactation | After birth | Lactation only | 21 days |
| C | Control | – | No supplementation | – |

During the testing days (the first four days), the pups were expected to find the platform within 60 seconds. If they did not find the platform within this time, the researcher assisted them in locating it. All pups that failed to find the platform were allowed to remain on the platform for 15 seconds to explore the cues. In the probe trial, the platform was removed, and the pups were expected to spend more time in the quadrant where the platform had been previously located, searching for the platform. There was a 30-minute interval between each trial, and each pup underwent a total of 17 trials over four days, with four trials each day and a probe trial on the fifth day. The pups were placed into the water by the same person, holding them by the nape of the neck with their faces facing the tank wall, ensuring that their tails and legs entered the water first. To avoid distracting the pups, the researcher wore the same lab coat throughout the experiment and did not use perfume or deodorant.

For evaluation, time was recorded using a digital stopwatch. During the testing days, the time taken to find the platform within 60 seconds was measured. If a pup found the platform in less than 60 seconds, it was considered successful, and the time was recorded; if not, it was considered unsuccessful. In the probe trial, the time spent in the quadrant where the platform had previously been located was calculated as a percentage of the total time spent in the tank (60 seconds), and the average percentage was determined for each group. Performance was assessed based on changes in the time taken to find the platform.

Statistical analyses were conducted using the Statistical Package for the Social Sciences (SPSS) version 25.0. The normality of the variables was assessed using the Shapiro-Wilk test. Descriptive analyses presented means ± standard deviations for quantitative variables. The "One-Way Analysis of Variance (ANOVA)" test was used to examine significant differences in quantitative variables between rat groups. Differences between more than two dependent measures (rat platform-finding times on different test days) were analyzed using the "Repeated Measures ANOVA." The "Two-Way Repeated Measures ANOVA" was employed to identify differences between repeated measures across rat groups. Tukey and Bonferroni multiple comparison tests were used to determine which specific groups contributed to significant differences. Results with p-values less than 0.05 were considered statistically significant.

## Results

The initial body weights, pre-mating weights, pre-birth weights, weight gained during, the number of offspring born to mothers and the gestation period by groups pregnancy of female rats are presented in Table 2. While there is a statistically significant difference in the mean initial weights, pre-mating weights, and pre-birth weights, number of offspring, gestation period (day) among the groups ($p = 0.000$, $p = 0.000$, $p = 0.001$, $p = 0.000$, $p = 0.000$ respectively), the mean weight gained during pregnancy does not show a statistically significant difference among the groups ($p = 0.372$).

When comparing initial weights, it was observed that the average weight of the preconception and 1st-week pregnancy groups was higher than that of the other groups, and this difference was statistically significant. When pre-delivery weight was examined, the difference was significant between the preconception group, the 1st- and 2nd-week pregnancy groups, and the 3rd-week pregnancy, lactating, and control groups.

The weight changes of the newborn rat pups are presented in Table 3. While there is a statistically significant difference among the groups in the mean weights on day 14, day 21, and day 45 after birth ($p = 0.000$, $p = 0.000$, $p = 0.000$, respectively), the mean weights on day 45 do not show a statistically significant difference among the groups ($p = 0.278$).

The average times for the rat pups to find the platform by day and based on the experimental group of their mothers are presented in Table 4. The average time on the first day was $25.06 \pm 1.54$ seconds, on the second day $20.76 \pm 1.28$ seconds, on the third day $19.14 \pm 1.22$ seconds, and on the fourth day $12.92 \pm 0.83$ seconds. The differences between the times are statistically significant ($p = 0.000$). The average times to find the platform on the 1st and 2nd experiment days were statistically significant ($p = 0.004$; $p = 0.002$, respectively), while on the 3rd and 4th days, no statistically significant difference was found ($p > 0.05$).

**Table 2. Descriptive Information of Mother Rats.**

| Groups | n | Initial Body Weights (g) X±SS | F p† | Pre-Mating Weights (g) X±SS | F p† | Pre-Birth Weights (g) X±SS | F p† | Weight Gained During Pregnancy (g) X±SS | p† | Number of Offspring X±SS | F p† | Gestation Period (day) X±SS | F p† |
|---|---|---|---|---|---|---|---|---|---|---|---|---|---|
| P | 5 | 303±11.35[a] | 13.797 0.000* | 325±14.14[c] | 29.976 0.000* | 479±38.35[e] | 4.850 0.001* | 154±34.22 | 0.372 | 16.4±2.27[a] | 5,506 0.000* | 20.00±4.85 | 7.623 0.000* |
| G1 | 5 | 311±48.23[b] | | 322±22.26[d] | | 487±37.80[f] | | 191±33.06 | | 14.4±1.42 | | 20.80±2.04[c] | |
| G2 | 5 | 287±9.19[a] | | 317±17.19[c] | | 462±35.53 | | 175±29.05 | | 15.0±3.26[b] | | 22.60±0.52[c,d] | |
| G3 | 5 | 249±12.65[a,b] | | 268±17.19[c,d] | | 418±60.97[e,f] | | 169±51.57 | | 13.8±1.55 | | 20.20±3.22[c] | |
| L | 5 | 263±7.15[a,b] | | 278±9.77[c,d] | | 426±15.42[f] | | 163±12.73 | | 14.0±2.40 | | 15.40±0.52[c] | |
| C | 5 | 259±7.00[a,b] | | 272±8.56[c,d] | | 420±62.18[f] | | 161±57.48 | | 11.6±1.07[a,b] | | 17.80±3.35 | |
| Total | 30 | 278±31.19 | | 297±28.95 | | 448±51.47 | | 168±39.39 | | 14.2±2.50 | | 19.46±3.60[d] | |

Post Hoc test, there is a significant difference between a, b, c, d, e, f groups. *p<0.05

**Table 3. The Weight Changes of The Newborn Rat Pups.**

| Groups | n | 14th. day (g) X±SS | F p† | 21st. day (g) X±SS | F p† | 30th. day (g) X±SS | F p† | 45th. day (g) X±SS | F p† |
|---|---|---|---|---|---|---|---|---|---|
| P | 10 | 25.05±2.87[a,b] | 10.316 0.000* | 44.11±6.14[c,d] | 22.572 0.000* | 49.25±9.43 | 0.278 | 95.25±13.81 | 2.704 0.030* |
| G1 | 10 | 25.89±2.51[b] | | 44.39±5.87[c,d] | | 52.50±4.41 | | 100.00±7.35 | |
| G2 | 10 | 24.98±4.11[a,b] | | 41.75±5.83[c,d] | | 47.80±8.20 | | 89.75±7.01 | |
| G3 | 10 | 26.93±2.10[b] | | 48.86±5.12[c,d] | | 51.75±3.91 | | 88.00±9.55[e] | |
| L | 10 | 29.88±2.07[a] | | 59.00±2.11[c] | | 54.25±4.74 | | 102.25±13.25 | |
| C | 10 | 32.98±4.35[b] | | 62.66±7.94[d] | | 51.25±4.29 | | 91.00±13.55[e] | |
| Total | 60 | 27.62±4.19 | | 50.13±9.71 | | 51.09±6.29 | | 94.37±11.91 | |

Post Hoc test, there is a significant difference between a, b, c, d groups. *p<0.05

The comparison of the times of the rat pups finding the platform according to the trial order on each trial day and their distribution according to the groups are given in Table 5. The average time of the first trials conducted each day was found to be 23.83±1.92 seconds; the average of the second trials was 19.88±1.34 seconds; the third trial averages were 19.13±0.87 seconds; and the average of the fourth trials was 15.32±0.94 seconds. The difference between the times is statistically significant (p=0.040). When the distribution by groups was examined, only the average difference between the third trials of the groups was found to be statistically significant (p=0.049).

The average percentage of time spent in the quadrant where the platform was previously located within 60 seconds during the probe test (Day 5) is presented in Table 6. The average percentage of time the pups spent in the quadrant where the platform was previously located was 28.50±11.23 seconds for the preconception group, 36.33±8.45 seconds for the 1st week of pregnancy, 37.50±8.65 seconds for the 2nd week of pregnancy, 30.66±11.36 seconds for the 3rd week of pregnancy, 34.00±13.38 seconds for the lactating group, and 37.66±9.59 seconds for the control group. There is no statistically significant difference was found between the group averages (p>0.05).

## Discussion

Maternal consumption of a PUFA-rich diet during pregnancy is critically important for the development of the fetal neurological system. Additionally, the idea that maternal PUFA intake has long-term effects on the offspring's body composition

**Table 4. The Average Times For The Rat Pups To Find The Platform By Day And Groups.**

| Day | n | X±SS (s) | | | | p† |
|---|---|---|---|---|---|---|
| 1st. day | 60 | 25.06±1.54a | | | | 0.000* |
| 2nd. day | 60 | 20.76±1.28b | | | | |
| 3rd. day | 60 | 19.14±1.22a,c | | | | |
| 4th. day | 60 | 12.92±0.83a,b,c | | | | |
| 1st. day | | Total Trial | Succesful Trial | Success Ratio (%) | | |
| P | 10 | 40 | 20 | 50 | 27.53±15.95d | 3.889 |
| G1 | 10 | 40 | 23 | 57.5 | 28.94±9.78e | 0.004* |
| G2 | 10 | 40 | 25 | 62.5 | 26.23±11.59 | |
| G3 | 10 | 40 | 12 | 30 | 9.99±9.98d,e,f,g | |
| L | 10 | 40 | 23 | 57.5 | 29.10±15.82f | |
| C | 10 | 40 | 24 | 60 | 25.06±13.35g | |
| 2nd. day | | | | | | |
| P | 10 | 40 | 32 | 80 | 18.85±9.24 | 4.540 |
| G1 | 10 | 40 | 32 | 80 | 16.08±10.86h | 0.002* |
| G2 | 10 | 40 | 27 | 67.5 | 11.76±7.38i | |
| G3 | 10 | 40 | 27 | 67.5 | 25.38±6.25 | |
| L | 10 | 40 | 30 | 75 | 22.05±6.25 | |
| C | 10 | 40 | 33 | 82.5 | 30.46±12.10h,i | |
| 3rd. day | | | | | | |
| P | 10 | 40 | 34 | 85 | 20.80±13.00 | 0.465 |
| G1 | 10 | 40 | 35 | 87.5 | 22.64±8.96 | |
| G2 | 10 | 40 | 33 | 82.5 | 17.51±6.92 | |
| G3 | 10 | 40 | 28 | 70 | 17.24±12.19 | |
| L | 10 | 40 | 31 | 77.5 | 15.23±7.81 | |
| C | 10 | 40 | 32 | 80 | 21.44±5.82 | |
| 4th. day | | | | | | |
| P | 10 | 40 | 37 | 92.5 | 11.29±6.00 | 0.113 |
| G1 | 10 | 40 | 38 | 95 | 14.90±5.61 | |
| G2 | 10 | 40 | 39 | 97.5 | 9.58±6.03 | |
| G3 | 10 | 40 | 30 | 75 | 15.76±8.10 | |
| L | 10 | 40 | 31 | 77.5 | 15.56±8.16 | |
| C | 10 | 40 | 33 | 82.5 | 10.45±3.42 | |

Post Hoc test, there is a significant difference between a, b, c, d, e, f, g, h, i groups. *p<0,05

and metabolic health has gained interest, in parallel with the growing popularity of the Developmental Origins of Health and Disease hypothesis. Periods referred to as sensitive windows of development, such as the intrauterine period, have a stronger impact on the child's future health status compared to other life stages [61]. This study is an in vivo investigation aimed at examining the effects of n-3 fatty acid supplementation, started in the preconception period, at different stages of pregnancy, and during lactation, on cognitive functions, and is considered unique in terms of exploring the impact of timing.

Since the rats were randomly assigned to groups at the beginning of the experiment, the difference in initial weights could not be controlled. Consequently, the difference in pre-mating weight also stems from these same groups. This resulted in a difference between the groups in terms of the amount of n-3 fatty acid supplementation consumed based

**Table 5. The Average Times For The Rat Pups To Find The Platform By Trial And Groups.**

| Day | n | X±SS (s) | | | | p† |
|---|---|---|---|---|---|---|
| 1st. trial | 60 | 23.83±1.92a | | | | 0.040 |
| 2nd. trial | 60 | 19.88±1.34 | | | | |
| 3rd. trial | 60 | 19.13±0.87b | | | | |
| 4th. trial | 60 | 15.32±0.94a,b | | | | |
| 1st. trial | | Total Trial | Succesful Trial | Success Ratio (%) | | |
| P | 10 | 40 | 17 | 42.5 | 22.13±19.27 | 0.341 |
| G1 | 10 | 40 | 20 | 50 | 30.45±13.86 | |
| G2 | 10 | 40 | 20 | 50 | 16.55±10.06 | |
| G3 | 10 | 40 | 16 | 40 | 23.12±17.05 | |
| L | 10 | 40 | 21 | 52.5 | 21.86±14.32 | |
| C | 10 | 40 | 27 | 67.5 | 28.88±13.38 | |
| 2nd. trial | | | | | | |
| P | 10 | 40 | 36 | 90 | 22.21±10.71 | 0.534 |
| G1 | 10 | 40 | 33 | 82.5 | 19.51±7.86 | |
| G2 | 10 | 40 | 31 | 77.5 | 15.81±9.47 | |
| G3 | 10 | 40 | 21 | 52.5 | 18.40±15.74 | |
| L | 10 | 40 | 27 | 67.5 | 19.00±5.39 | |
| C | 10 | 40 | 26 | 65 | 24.34±10.32 | |
| 3rd. trial | | | | | | |
| P | 10 | 40 | 34 | 85 | 13.49±5.56c | 0.049* |
| G1 | 10 | 40 | 38 | 95 | 17.80±10.10 | |
| G2 | 10 | 40 | 36 | 90 | 18.54±6.35 | |
| G3 | 10 | 40 | 32 | 80 | 20.64±4.50 | |
| L | 10 | 40 | 33 | 82.5 | 22.40±5.57c | |
| C | 10 | 40 | 34 | 85 | 21.94±7.03 | |
| 4th. trial | | | | | | |
| P | 10 | 40 | 36 | 90 | 14.24±6.41 | 0.608 |
| G1 | 10 | 40 | 37 | 92.5 | 18.10±8.59 | |
| G2 | 10 | 40 | 37 | 92.5 | 12.85±4.69 | |
| G3 | 10 | 40 | 28 | 70 | 14.92±9.32 | |
| L | 10 | 40 | 33 | 82.5 | 14.58±5.89 | |
| C | 10 | 40 | 35 | 87.5 | 17.22±7.77 | |

Post Hoc test, there is a significant difference between a, b, c groups. *p<0.05

on body weight. Due to the n-3 fatty acid supplementation started before mating, the addition of energy from fat to the ad libitum diet resulted in a greater weight gain in the preconception group compared to the other groups, which was an expected outcome. The higher pre-delivery weight in the group that received n-3 fatty acid supplementation earlier in pregnancy compared to those that received it for a shorter period can be explained by the increase in the number of offspring and the longer duration of added energy from fat in the diet. In terms of the number of offspring, the control group had the fewest. The significance of the difference in the number of offspring between groups is attributed to the preconception and 2nd-week pregnancy groups compared to the control group. The group with the shortest gestation period was the lactating group. The significance of the difference in gestation period comes from the difference between the lactating group and the 1st-, 2nd-, and 3rd-week pregnancy groups, as well as the control group and the 2nd-week pregnancy group.

**Table 6. Probe Test Statistics.**

| Groups | n | X±SS (%) | p† |
|---|---|---|---|
| P | 10 | 28.50±11.23 | 0.306 |
| G1 | 10 | 36.33±8.45 | |
| G2 | 10 | 37.50±8.65 | |
| G3 | 10 | 30.66±11.36 | |
| L | 10 | 34.00±13.38 | |
| C | 10 | 37.66±9.59 | |
| Total | 60 | 34.11±10.72 | |

In a study conducted on humans, from the 14th week of pregnancy until the 32nd week postpartum, margarine containing ALA+LA and margarine containing only LA were consumed. Cognitive function tests were applied at weeks 14, 17, 29, and 36 of pregnancy and at 32 weeks postpartum. It was found that different tests conducted at different times did not create a statistically significant difference [62]. Although ALA is from the PUFA family, its inability to produce the expected effect suggests that EPA and DHA, which were also used in this study, have a greater impact on cognitive function.

In another study conducted on rats fed a semi-synthetic diet containing 5% olive oil or fish oil during pregnancy, the group consuming the fish oil diet gained less weight during pregnancy compared to those consuming olive oil, but the number of offspring was higher in the fish oil group [56]. In another study investigating the effects of n-3 fatty acids, vitamin B12, and folic acid given during pregnancy, it was stated that the groups were similar in terms of weight gained during pregnancy and offspring weight [12]. In a separate study where n-3 fatty acid supplementation was given to female rats before pregnancy, the pre-mating and pre-delivery weights were higher in the control group compared to the n-3 fatty acid group, while weight gained during pregnancy was lower in the n-3 fatty acid group [59]

In this study, found that the pups' performance improved day by day, with a statistically significant reduction in the time taken to find the platform (p<0.05). The difference between the groups stemmed from the comparison of Days 1 and 3 and from Day 4 compared to all other days. On the last day (Day 4), the shortest average time to find the platform was observed, indicating that all groups had learned. The difference between the groups on the first and second days was statistically significant (p<0.05). On the first day, the average for the 3rd-week pregnancy group was lower than the other groups, while on the second day, the 2nd-week pregnancy group had the lowest average. However, in terms of success percentages, the lowest success rate on Day 1 was observed in the 3rd-week pregnancy group, and on Day 2, it was observed in the 2nd- and 3rd-week pregnancy groups. When the differences between trials were examined, the difference between the first and last trials on the same day was statistically significant (p<0.05). The significant difference between the groups was attributed to the third trial, specifically between the preconception and lactating groups (p<0.05).

In a study by Kavraal et al. on the offspring of mother rats receiving n-3 fatty acid supplementation from the beginning of pregnancy, it was found that the time spent in the Morris Water Maze by all pups decreased over consecutive days, showing an improvement in performance. The time taken to find the platform on trial days and the probe trial was not statistically significant between the offspring of the omega-3 fatty acid supplementation group and the placebo group. However, the time taken to find the platform on the first and second days indicated better learning performance in the n-3 fatty acid supplementation group [52]. In another study, where rats were given a diet deficient in or normal in n-3 fatty acids during pregnancy and lactation, and the pups were given either water as a placebo or n-3 fatty acid supplementation after weaning, spatial and working memory were tested using the Morris Water Maze. The pups of mothers on a normal diet who received n-3 fatty acid supplementation showed better results compared to the other groups [58]. In a study examining whether nicotine exposure during pregnancy and lactation could be reduced with n-3 fatty acid supplementation, it was found that the time taken to find the platform decreased over days in both the control group and the n-3 fatty acid

supplementation group, but there was no significant difference between the two groups. Similarly, no statistically significant difference was found in the probe trial between the n-3 fatty acid group and the control or sham groups [63].

Cognitive function tests (Open Field Test, Elevated Plus-Maze, and Barnes Circular Maze) conducted on the offspring of rats that received adequate or inadequate n-3 fatty acid supplementation during pregnancy found a statistically significant difference in long-term memory tests among the offspring of mothers who received sufficient n-3 fatty acid supplementation [57]. Success percentages, it was observed that the preconception, 1st-week, and 2nd-week pregnancy groups, which had longer exposure to n-3 fatty acid supplementation, found the platform at higher rates. It was concluded that long-term n-3 fatty acid supplementation could be associated with better cognitive function. No statistically significant difference was found between the groups in the probe test, which measures reference memory. This may be due to the possibility that a single test may not be sufficient to assess reference memory.

A limitation of this study is the absence of molecular assessments, such as measurements of brain- derived neurotropic factor (BDNF) protein levels and phosphorylated Cyclic AMP-Response Element Binding Protein (CREB) expression, which could have provided further insight into the neurobiological mechanisms underlying the observed cognitive outcomes. Although these molecular markers are valuable for elucidating the pathways through which omega-3 supplementation may affect brain function, our study was designed to focus on behavioral endpoints. We recommend that future research consider incorporating these molecular evaluations to achieve a more comprehensive understanding of the mechanisms involved.

## Conclusion

The preconception period and the first 1,000 days of life are critical for shaping future generations' health and cognitive potential, which in turn impacts societal development. Planned pregnancies, healthy lifestyle choices, and addressing nutritional needs—especially during pregnancy and lactation—are essential. Where dietary intake is insufficient, appropriate supplementation should be considered. Even small improvements at the individual level can yield significant public health benefits and support the formation of healthier societies.

## Author contributions

**Conceptualization:** Hande Nur Onur Öztürk, Perim Fatma Türker.

**Data curation:** Hande Nur Onur Öztürk.

**Formal analysis:** Hande Nur Onur Öztürk.

**Methodology:** Perim Fatma Türker.

**Project administration:** Perim Fatma Türker.

**Supervision:** Perim Fatma Türker.

**Writing – original draft:** Hande Nur Onur Öztürk.

**Writing – review & editing:** Hande Nur Onur Öztürk.

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
