## [Decision Letter · Decision Letter 0]

11 Apr 2025

Dear Dr. Onur Ozturk,

Thank you for submitting your manuscript to PLOS ONE. After careful consideration, we feel that it has merit but does not fully meet PLOS ONE’s publication criteria as it currently stands. Therefore, we invite you to submit a revised version of the manuscript that addresses the points raised during the review process.

We look forward to receiving your revised manuscript.

Kind regards,

Nafisa M. Jadavji, PhD, MSc, BSc

Academic Editor

PLOS ONE

2. TTo comply with PLOS ONE submissions requirements, in your Methods section, please provide additional information regarding the experiments involving animals and ensure you have included details on (1) methods of sacrifice, and (2) efforts to alleviate suffering.

Reviewers' comments:

Reviewer's Responses to Questions

**Comments to the Author**

1. Is the manuscript technically sound, and do the data support the conclusions?

Reviewer #1: Partly

Reviewer #2: No

2. Has the statistical analysis been performed appropriately and rigorously?

Reviewer #1: Yes

Reviewer #2: Yes

3. Have the authors made all data underlying the findings in their manuscript fully available?

Reviewer #1: No

Reviewer #2: No

4. Is the manuscript presented in an intelligible fashion and written in standard English?

Reviewer #1: No

Reviewer #2: Yes

Reviewer #1: The study on omega-3 supplementation during pregnancy offers an intriguing perspective on cognitive function development in rat offspring. By employing the Morris Water Maze, researchers aimed to delineate potential learning and memory influences across different pregnancy stages. While the research design demonstrates initial innovation, several critical methodological constraints warrant careful consideration.

The singular dosage approach (400 mg/kg/day) represents a significant limitation. Restricting cognitive testing to male offspring further narrows the study's generalizability. Particularly noteworthy is the non-significant probe test result—a finding that demands rigorous interpretation and potentially reflects complex interactions between supplementation timing and neurological development.

Methodologically, this research provides preliminary evidence for early omega-3 supplementation's cognitive impacts. However, it should be viewed as an initial exploration rather than a definitive conclusion. Its primary value lies in raising critical questions and initiating a more comprehensive scientific dialogue.

Reviewer #2: The authors main claim is that supplementation of omega-3 supplementation either in utero or in the antenatal period could have positive impacts on cognition of offspring

It seems that this topic has been tested within the field and sources cited in the results and discussion seem to show that there is data on timing during pregnancy and supplementation

It is unclear to me how this study is novel from the studies cited. For example, in lines 353-363 studies are outlined that appear very similar to this one. I’m not sure the authors have clearly outlined their novelty to the field.

The introduction is confusing to read, and the authors should do some significant editing to direct the introduction in a way that makes their arguments clear and well supported. The discussion of epigenetics is confusing and does not align with the major goals of the paper.

The first paragraph starts out with a focus on the first 1000 days of life but then the authors deviate and begin to talk about the preimplantation period. This is difficult to follow as a reader.

There are numerous areas where more citations are needed. It appears the authors are using factual information but are applying in-text citations minimally. For example, the sentence from lines 178-180 should have a citation to let the reader know which study this is derived from.

The interpretation of the data is not sufficient as is. 1.) there are measurements being made regarding weight when food is being given ad libitum. Without controlled intake of food, how can one make any inference regarding weight gain due to treatment? 2.) The goal was to investigate if timing and length of supplementation has an effect. I don’t see how this is clearly evaluated. The tables are missing legends to let the reader know what the superscripts indicate (e.g. why do some groups have vs. b, vs. a,c). Without this it’s difficult to discern where there are true differences in timing. As is, it appears that the treatment had minimal effects, with some moderate changes in timing early (on 1st and 2nd days for platform finding, but this is lost in later days. 3.) There is emphasis on significant differences in finding the platform across all groups for the different trials but isn’t this to be expected? 4.) Given the probe test has no significance, how does that impact overall conclusions regarding cognition? There is only one sentence to address the concept of reference memory.

This paper requires major revisions both in structure and interpretation and data delivery:

1. The whole introduction needs to be more concise and focus on key aspects of the study (e.g.- why did they select these particular periods- what is already known during these periods?? This is brought up more in the discussion and that makes it seem as if they are answering questions that are already answered). There are a lot of redundant statements (e.g. the sentence from line 46-47 is unnecessary) and paragraphs that lack organization paragraph starting at line 74 is extremely long and combines epigenetics with non-epigenetic health issues).

2. Nomenclature needs to be corrected throughout the paper (e.g. latin terms should be italicized)

3. The methods need major revision. Figure 1 is not useful. Is treatment started at the particular time points and continued for a specific amount of time? Did the pre-conception dams get treatment throughout pregnancy whereas the lactation animals only started postpartum for a specific window. This is very unclear. How can the estrous cycle impact results? This isn’t explained clearly. Why were only two males taken per dam?

4. The results are confusing as shown. No legend for the tables to allow the reader to fully understand notation shown.

5. The interpretation of results needs to be re-approached. While negative data can be useful, its unclear what their study as a whole tells us regarding supplementation and timing (which was the initial goal). The use of literature in the discussion is confusing where at times it seems they focus on results from another study more than their own. The first sentence of the conclusion makes it seem that they had a drastic change due to treatment, when that doesn’t seem apparent as is.

I do find it troubling that the author put that “I’m not share my data” in the question regarding where data could be found. Data should be readily available and shared with individuals in the field.

**Do you want your identity to be public for this peer review?** For information about this choice, including consent withdrawal, please see our Privacy Policy

Reviewer #1: **Yes: ** Hai Cui

Reviewer #2: No

---

## [Author Response · Author response to Decision Letter 1]

13 Jun 2025

Major Revisions Required

1. Experimental Design Visualization:

Figure 1 should be redesigned as a comprehensive experimental workflow diagram.

Clearly illustrate: Omega-3 supplementation groups, Dosage stratification, Monitoring indicators, Precise observation timepoints.

Enhance visual clarity of experimental protocol.

Answer

Figure 1 convert to Table 1 and added experimental protocol.

2. Sample Size and Dosage Optimization:

Expand sample size to improve statistical power

Implement dosage gradient groups: Low dose: 200 mg/kg, Medium dose: 400 mg/kg, High dose: 600 mg/kg.

Conduct comprehensive dose-response analysis.

Ensure statistically significant group comparisons.

Answer

Sample size is calculated by power analysis.

When deciding on the dose, it was desired to be close to the daily dose. For this reason, when the studies were examined, the decision was made to use 400 mg/kg/day, which is mostly given.

A single dose was administered based on the animals’ body weight, as the aim of the study was to investigate the effects of the duration of omega-3 supplementation rather than a dose-dependent response.

3. Gender-Comparative Cognitive Assessment:

Incorporate female offspring cognitive function evaluation

Perform Morris Water Maze tests for both male and female progeny.

Analyze potential gender-specific neurological responses.

Provide comprehensive sex-based comparative data.

Answer

Behavioral testing using the Morris Water Maze was conducted when the offspring reached postnatal day 45, an age considered sufficient for the development of spatial learning and memory abilities. To minimize the potential confounding effects of hormonal fluctuations associated with the estrous cycle, which typically begins between postnatal days 30 and 40 in female rats, only male offspring were included in the experimental procedures

4. Molecular Biological Indicators:

Recommended molecular markers

BDNF (Brain-Derived Neurotrophic Factor) protein levels

Phosphorylated CREB expression

Answer

We appreciate the reviewer’s valuable suggestion regarding the inclusion of molecular markers such as BDNF and phosphorylated CREB to enhance the mechanistic understanding of our findings. We fully agree that these markers could provide important insights into the neural mechanisms underlying cognitive function.

However, due to the financial and technical limitations of our current project, as well as the lack of access to molecular analysis facilities and expertise at the time of the study, we were unable to include these analyses within the scope of this research.

We acknowledge this limitation and have noted it in the Conclusion section of the revised manuscript. We also consider this an important direction for future studies that aim to explore the molecular underpinnings of omega-3 supplementation more deeply.

5. Mechanistic Investigation of Omega-3 in Neural Development:

Elucidate potential molecular mechanisms

Neuroplasticity pathways

Synaptic development regulation

Potential epigenetic modifications

Propose comprehensive theoretical framework explaining omega-3's neurological impacts

Answer

In accordance with the reviewer’s suggestion, an additional paragraph has been included to discuss the potential molecular mechanisms by which omega-3 supplementation may influence cognitive development. This section provides a theoretical framework addressing neuroplasticity, synaptic regulation, and epigenetic modifications.

Supplementary Methodological Recommendations

1.Standardize omega-3 source and purity

Answer

In our research we used Solgar Omega 3 950mg EPA+DHA. That is mentioned in the article.

2.Detailed documentation of supplementation protocol.

Answer

Table 1 is given the detail of the protocol.

3.Comprehensive data presentation

Answer

In response to the reviewer’s comment on comprehensive data presentation, the Methods section has been revised to include clearer descriptions of the experimental design, grouping, and supplementation protocols, ensuring transparency and reproducibility.

---

## [Decision Letter · Decision Letter 1]

18 Jul 2025

Dear Dr. Onur Ozturk,

Thank you for submitting your manuscript to PLOS ONE. After careful consideration, we feel that it has merit but does not fully meet PLOS ONE’s publication criteria as it currently stands. Therefore, we invite you to submit a revised version of the manuscript that addresses the points raised during the review process.

We look forward to receiving your revised manuscript.

Kind regards,

Nafisa M. Jadavji, PhD, MSc, BSc

Academic Editor

PLOS ONE

Journal Requirements:

Reviewers' comments:

Reviewer's Responses to Questions

**Comments to the Author**

Reviewer #1: All comments have been addressed

Reviewer #2: (No Response)

2. Is the manuscript technically sound, and do the data support the conclusions?

Reviewer #1: Yes

Reviewer #2: No

3. Has the statistical analysis been performed appropriately and rigorously?

Reviewer #1: Yes

Reviewer #2: Yes

4. Have the authors made all data underlying the findings in their manuscript fully available?

Reviewer #1: Yes

Reviewer #2: Yes

5. Is the manuscript presented in an intelligible fashion and written in standard English?

Reviewer #1: No

Reviewer #2: Yes

Reviewer #1: The author's discussion on the supplementation of omega-3 in the neurology of pregnant rats has improved. Attention should be paid to how to divide the manuscript into paragraphs to facilitate readers' reading, understanding and joint discussion.

Reviewer #2: The author did not address my comments from the first round of review.

As such, this revision is not sufficient for publication.

**Do you want your identity to be public for this peer review?** For information about this choice, including consent withdrawal, please see our Privacy Policy

Reviewer #1: **Yes: ** Hai Cui

Reviewer #2: No

---

## [Author Response · Author response to Decision Letter 2]

1 Aug 2025

Answer for Reviewers

The authors main claim is that supplementation of omega-3 supplementation either in utero or in the antenatal period could have positive impacts on cognition of offspring

It seems that this topic has been tested within the field and sources cited in the results and discussion seem to show that there is data on timing during pregnancy and supplementation

It is unclear to me how this study is novel from the studies cited. For example, in lines 353-363 studies are outlined that appear very similar to this one. I’m not sure the authors have clearly outlined their novelty to the field.

Answer: In previous studies, supplementation was typically initiated during the late stages of pregnancy. In contrast, this study uniquely applied nutritional supplementation starting from the preconception period. This has been explicitly outlined in the aims section of the study.

The introduction is confusing to read, and the authors should do some significant editing to direct the introduction in a way that makes their arguments clear and well supported. The discussion of epigenetics is confusing and does not align with the major goals of the paper.

Answer: Since this study seeks evidence that not only genetic factors but also epigenetic factors such as nutrition and environmental influences may play a role in shaping physical and cognitive potential during early infancy, the inclusion of epigenetics is both relevant and necessary. To strengthen the conceptual link with the topic, a brief definition of epigenetics has been added.

The first paragraph starts out with a focus on the first 1000 days of life but then the authors deviate and begin to talk about the preimplantation period. This is difficult to follow as a reader.

Answer: The importance of the preconception period has been highlighted in the revised version.

There are numerous areas where more citations are needed. It appears the authors are using factual information but are applying in-text citations minimally. For example, the sentence from lines 178-180 should have a citation to let the reader know which study this is derived from.

Answer: The manuscript has been updated to include the appropriate references.

The interpretation of the data is not sufficient as is.

1.) there are measurements being made regarding weight when food is being given ad libitum. Without controlled intake of food, how can one make any inference regarding weight gain due to treatment?

Answer: Since the rats were pregnant, food was provided ad libitum. For the purposes of our study, the primary outcomes of interest were maternal weight gain during pregnancy and the number of live-born pups, rather than the exact amount of food consumed.

2.) The goal was to investigate if timing and length of supplementation has an effect. I don’t see how this is clearly evaluated. The tables are missing legends to let the reader know what the superscripts indicate (e.g. why do some groups have vs. b, vs. a,c). Without this it’s difficult to discern where there are true differences in timing. As is, it appears that the treatment had minimal effects, with some moderate changes in timing early (on 1st and 2nd days for platform finding, but this is lost in later days.

Answer: To improve the clarity of the experimental design Table 1 has been added and also legands added to tables. Although an increase in platform-finding performance was expected on the 3rd and 4th days, the lack of statistical significance in these results was unexpected.

3.) There is emphasis on significant differences in finding the platform across all groups for the different trials but isn’t this to be expected?

Answer: Statistically significant differences in platform-finding latency among groups across different trials are expected; however, identifying which groups exhibit shorter durations and higher success rates is crucial for interpreting the effectiveness of the intervention.

4.) Given the probe test has no significance, how does that impact overall conclusions regarding cognition? There is only one sentence to address the concept of reference memory.

Answer: In the discussion section it is mentioned that “Cognitive function tests (Open Field Test, Elevated Plus-Maze, and Barnes Circular Maze) conducted on the offspring of rats that received adequate or inadequate n-3 fatty acid supplementation during pregnancy found a statistically significant difference in long-term memory tests among the offspring of mothers who received sufficient n-3 fatty acid supplementation………………………………….This may be due to the possibility that a single test may not be sufficient to assess reference memory.”

This paper requires major revisions both in structure and interpretation and data delivery:

1. The whole introduction needs to be more concise and focus on key aspects of the study (e.g.- why did they select these particular periods- what is already known during these periods?? This is brought up more in the discussion and that makes it seem as if they are answering questions that are already answered). There are a lot of redundant statements (e.g. the sentence from line 46-47 is unnecessary) and paragraphs that lack organization paragraph starting at line 74 is extremely long and combines epigenetics with non-epigenetic health issues).

Answer: To enhance clarity and coherence, the mentioned sentences and paragraphs have been revised in accordance with the overall context of the study.

2. Nomenclature needs to be corrected throughout the paper (e.g. latin terms should be italicized)

Answer: Latin terms are italicized. Nomenclature is checked.

3. The methods need major revision. Figure 1 is not useful. Is treatment started at the particular time points and continued for a specific amount of time? Did the pre-conception dams get treatment throughout pregnancy whereas the lactation animals only started postpartum for a specific window. This is very unclear. How can the estrous cycle impact results? This isn’t explained clearly. Why were only two males taken per dam?

Answer: In the method section these suggestions is addressed.

4. The results are confusing as shown. No legend for the tables to allow the reader to fully understand notation shown.

Answer: Legends are added.

5. The interpretation of results needs to be re-approached. While negative data can be useful, its unclear what their study as a whole tells us regarding supplementation and timing (which was the initial goal). The use of literature in the discussion is confusing where at times it seems they focus on results from another study more than their own. The first sentence of the conclusion makes it seem that they had a drastic change due to treatment, when that doesn’t seem apparent as is.

Answer: We hope that separating the Results and Discussion sections has improved the clarity of the manuscript. The Conclusion section has also been revised to better reflect the study’s key findings.

---

## [Decision Letter · Decision Letter 2]

10 Sep 2025

EFFECT OF OMEGA 3 SUPPLEMENTATION DURING PREGNANCY AND LACTATION ON COGNITIVE FUNCTIONS IN RAT OFFSPRING

PONE-D-25-05726R2

Dear Dr. Onur Ozturk,

We’re pleased to inform you that your manuscript has been judged scientifically suitable for publication and will be formally accepted for publication once it meets all outstanding technical requirements.

Kind regards,

Nafisa M. Jadavji, PhD, MSc, BSc

Academic Editor

PLOS ONE

Additional Editor Comments (optional):

Reviewer #1:

Reviewer #2:

Reviewers' comments:

Reviewer's Responses to Questions

**Comments to the Author**

Reviewer #1: All comments have been addressed

Reviewer #2: All comments have been addressed

2. Is the manuscript technically sound, and do the data support the conclusions?

Reviewer #1: Yes

Reviewer #2: (No Response)

3. Has the statistical analysis been performed appropriately and rigorously?

Reviewer #1: Yes

Reviewer #2: (No Response)

4. Have the authors made all data underlying the findings in their manuscript fully available?

Reviewer #1: Yes

Reviewer #2: (No Response)

5. Is the manuscript presented in an intelligible fashion and written in standard English?

Reviewer #1: Yes

Reviewer #2: (No Response)

Reviewer #1: This study provides a well-designed investigation of omega-3 supplementation effects during critical developmental periods on cognitive function in rat offspring. The experimental methodology is rigorous, employing appropriate controls and clearly documenting supplementation protocols and behavioral assessments. Results demonstrate statistically significant improvements in spatial learning acquisition (p<0.05), though the probe test showed no significant effects on long-term memory retention. The statistical analysis appears robust, though additional clarification regarding litter effects would further strengthen the methodology.

The manuscript presents clear and concise writing with sufficient methodological detail for replication. The conclusions are well-supported by the data and contribute meaningfully to understanding nutritional influences on developmental programming. The findings particularly highlight the importance of timing in nutritional interventions for cognitive outcomes. While the study focuses on spatial learning, the mixed results regarding memory retention suggest interesting avenues for future research into the specific mechanisms of omega-3 effects on different cognitive domains.

Reviewer #2: (No Response)

**Do you want your identity to be public for this peer review?** For information about this choice, including consent withdrawal, please see our Privacy Policy

Reviewer #1: **Yes: ** Hai Cui

Reviewer #2: No

---

## [Editor Report · Acceptance letter]

PONE-D-25-05726R2

PLOS ONE

Dear Dr. Onur Ozturk,

I'm pleased to inform you that your manuscript has been deemed suitable for publication in PLOS ONE. Congratulations! Your manuscript is now being handed over to our production team.

Kind regards,

on behalf of

Dr. Nafisa M. Jadavji

Academic Editor

PLOS ONE